# Lignocellulosic-Based Activated Carbon-Loaded Silver Nanoparticles and Chitosan for Efficient Removal of Cadmium and Optimization Using Response Surface Methodology

**DOI:** 10.3390/ma15248901

**Published:** 2022-12-13

**Authors:** Sujata Mandal, Sreekar B. Marpu, Mohammad A. Omary, Catalin C. Dinulescu, Victor Prybutok, Sheldon Q. Shi

**Affiliations:** 1Ingram School of Engineering, Texas State University, San Marcos, TX 78666, USA; 2Department of Chemistry, University of North Texas, Denton, TX 76207, USA; 3College of Business, Tarleton State University, Stephenville, TX 76401, USA; 4Toulouse Graduate School, University of North Texas, Denton, TX 76201, USA; 5G. Brint Ryan College of Business, University of North Texas, Denton, TX 76201, USA; 6Department of Mechanical Engineering, University of North Texas, Denton, TX 76207, USA

**Keywords:** nano-adsorbent, kenaf, chitosan, cadmium, Box–Behnken design

## Abstract

The cadmium-contaminated water body is a worldwide concern for the environment and toxic to human beings and the removal of cadmium ions from drinking and groundwater sustainably and cost-effectively is important. A novel nano-biocomposite was obtained by impregnating silver nanoparticles (AgNPs) within kenaf-based activated carbon (KAC) in the presence of chitosan matrix (CS) by a simple, facile photoirradiation method. The nano-biocomposite (CS-KAC-Ag) was characterized by an environmental scanning electron microscope equipped with energy dispersive X-ray spectroscopy (ESEM-EDX), Fourier-transform infrared spectroscopy (FTIR), and Brunauer–Emmett–Teller (BET) method. A Box–Behnken design of response surface methodology (RSM) was used to optimize the adsorption of Cd^2+^. It was found that 95.1% of Cd^2+^ (10 mg L^−1^) was eliminated at pH 9, contact time of 120 min, and adsorbent dosage of 20 mg, respectively. The adsorption of Cd^2+^ by CS-KAC-Ag is also in agreement with the pseudo-second-order kinetic model with an R^2^ (coefficient of determination) factor greater than 99%. The lab data were also corroborated by tests conducted using water samples collected from mining sites in Mexico. Along with Cd^2+^, the CS-KAC-Ag exhibited superior removal efficiency towards Cr^6+^ (91.7%) > Ni^2+^ (84.4%) > Co^2+^ (80.5%) at pH 6.5 and 0.2 g L^−1^ dose of the nano-adsorbent. Moreover, the adsorbent was regenerated, and the adsorption capacity remained unaltered after five successive cycles. The results showed that synthesized CS-KAC-Ag was a biocompatible and versatile porous filtering material for the decontamination of different toxic metal ions.

## 1. Introduction

Water pollution is a critical issue around the world that threatens human health and the environment. The Integrated Risk Information System of the Environmental Protection Agency (EPA) has identified cadmium as a Group B1 potential human carcinogen, and the metals get into the environment through mining activities, smelting of iron, lead, or copper, and electroplating [1]. Long-term inhalation or exposure to the metal tends to build up cadmium in the kidney and can cause renal failure [2]. It is also responsible for bone diseases such as osteoporosis and a high rate of fractures [3,4]. The safety standards by the EPA recommend 0.005 mg L^−1^ or below as an acceptable level of cadmium in potable water [5]. 

Several studies suggested that adsorption is one of the most effective and widely favored technologies for water purification because of its high effectiveness, simplicity of operation, low cost, and limited dependency on complicated support infrastructure. Moreover, no additional toxic by-products are added to the environment [6,7]. Various conventional (e.g., activated carbons, zeolites, clays, bio-sorbents) and nano-structured absorbents (e.g., fullerenes, carbon nanotubes, and graphene) could remove bio-resilient and lethal heavy-metal ions from wastewater [8,9,10]. Recently, many agricultural lignocellulosic by-products were utilized to develop activated carbons (AC) that are low-cost and effective absorbents that provide a large surface area [11,12]. Recent research on AC-based and biochar-supported metallic nanomaterials has attracted considerable attention for the adsorption of heavy metals from water [13,14]. The use of polymer-supported silver nanoparticles for water purification has also been reported [15]. Recently, silver nanoparticles (AgNPs) loaded commercial AC matrix was used for the elimination of Cr^6+^ from water and loading of AgNPs on AC exhibited superior adsorption efficacy compared to pristine AC [16]. Kenaf (*Hibiscus cannabinus*), a natural cellulosic fiber comprised of cellulose, hemicellulose, and lignin and comprised of functional groups such as phenols, amines, and hydroxyls, plays a significant role in the adsorption of heavy metals and dyes [17,18]. Biopolymers are also applied currently for water treatment and chitosan (CS) is a non-toxic, inexpensive, and biocompatible biopolymer that is useful in the removal of metal ions from wastewater due to its abundant hydrophilic functional groups. Low chemical stability, porosity, and small surface area are some of the weaknesses which limit the performance of this natural material in the adsorption process [19]. Hence, the modification of CS is essential for the removal of effluents [20]. Chitosan easily forms complexes with metallic nanoparticles to form nanocomposites because of its ability to conjugate with metals through -NH_2_ group and this process has been widely applied for heavy metal remediation [21]. 

From the available literature, very few researchers have used kenaf-based activated carbon (KAC) mediated silver nanoparticles (AgNPs) and chitosan (CS)-based nano-adsorbent for the decontamination of mixed toxic metals from wastewater. To lessen the waste in the environment, the study used natural biomass. Chitosan (CS), a natural polymer, was used to enhance the stability of AgNPs. Cadmium was selected as a typical heavy metal as a sorbate owing to its high toxicity and prevalent industrial discharge. Thus, the novelty of this study focuses on the synthesis of a benign nanocomposite (CS-KAC-Ag) by facile, simple, and fast method by ‘self-activation’ from kenaf biomass as a precursor. Silver nanoparticles (AgNPs) are further immobilized in KAC using a biopolymer CS. Moreover, the fabricated novel nanobiocomposite exhibits unique properties with varied functional groups and is effective in removing Cd^2+^ and mixed metals Co^2+^, Cr^6+^, and Ni^2+^ from mining wastewater and capable of removing higher percentage of toxic metals than commercial AC using minimum adsorbent dosage. The traditional method to determine the optimum adsorption conditions at different operating parameters takes much effort and time. Hence, this study uses the Box–Behnken design (BBD) which suits the response surface methodology (RSM) to evaluate and understand the optimization process. This design of experiments uses statistical methods to ensure the validity and generalizability of the conclusions that can be drawn from much reduced and compare the removal efficiency of the nano-biocomposite with commercial activated carbon (CAC) and kenaf-based activated carbon (KAC). Furthermore, the study also highlights that CS-KAC-Ag successfully could be recycled for four successive adsorption–desorption cycles indicating its high reusability.

## 2. Materials and Methods

The kenaf core was obtained from “Kengro Corporation” (Charleston, MS, USA). Cadmium chloride hydrate (CdCl_2_ 98%), chitosan (CS) (degree of deacetylation: 75–85%, Mwt: 190,000–310,000), silver nitrate (AgNO_3_ ≥ 99.0%), sodium hydroxide (NaOH ≥ 97.0%, pellets), hydrochloric acid (HCl 37%), and nitric acid (HNO_3_ 70%) were purchased from Sigma-Aldrich (St. Louis, MO, USA). 

### 2.1. Preparation of the CS-KAC-Ag Nano-Absorbent

The KAC was prepared through a “self-activation” described in our previous work [22]. Chitosan powder was dissolved in 2 *v*/*v*% acetic acid solution to obtain 1% CS solution followed by stirring and dialysis. Then, 0.1% of CS solution was added to 1 mL of 1 M of ammonium hydroxide solution until the pH was adjusted between 6.5 and 7.0. Next, 1 mL of 1 mM AgNO_3_ and 50 mg of KAC were added to the CS solution. The solution was thoroughly mixed and homogenized, followed by irradiation (photo irradiated with magnetic stirring) using a spot curing machine (Lesco UV, Torrance, CA, USA, American Ultraviolet Spectrum100-LESCO) for 40 min. A schematic representation of the formation of silver nanoparticles within CS and KAC is shown in Figure 1. The reduction of the Ag^+^ ions was monitored by periodic sampling of 1 mL of the aliquots using a UV-Vis spectrophotometer (PerkinElmer Ltd., Shelton, CT, USA, Model No: Perkin-Elmer Lambda-900) at 15 min, 30 min, and 40 min. The CS-KAC-Ag was washed, and oven dried for 6–7 h at 40 °C. Then, the dried nano-adsorbent was kept in a desiccator for further use. 

### 2.2. Characterization of the Nano-Adsorbent

Environmental scanning electron (FEI Company, Oregon, USA, ESEM-EDAX, ESEM FEI Quanta 200) was used to investigate the surface morphology and composition of the raw biomass kenaf, KAC, and CS-KAC-Ag before and after uses. The surface area, pores volume, and average pore size of the KF, KAC, and CS-KAC-Ag were examined by the Brunauer–Emmett–Teller (BET) method (N_2_ adsorption) using a surface area analyzer (3Flex 3500 from Micromeritics Instrument Corporation, Norcross, GA, USA). The samples were degassed at 150 °C before the N_2_ adsorption. Fourier-transform infrared radiation (FTIR Spectrometer Spectrum Two, Perkin Elmer Inc., Waltham, MA, USA) was used to analyze the chemical content of the KF, KAC, and CS-KAC-Ag samples.

### 2.3. Adsorbate Cd^2+^

All chemicals used were of analytical grade. Cadmium chloride was used to prepare 1000 mg L^−1^ of Cd^2+^ standard stock solution. The stock solution is diluted with ultra-pure Millipore water for different dilutions. A total of 1 M HCl and 1 M NaOH were used for maintaining pH concentration.

## 3. Experiment

### 3.1. Adsorption Studies

Adsorption kinetics of the fabricated nanobiocomposite were studied using aqueous solutions of the Cd ^2+^ metal ions. By diluting the stock solutions, standard metal ions concentrations of 10 mg L^−1^, 30 mg L^−1^, and 50 g L^−1^ were prepared. A minimum adsorbent dosage of 20 mg was added in 100 mL using 10 mg L^−1^ of Cd^2+^, and pH~6.0. For conducting the adsorption experiment, the flasks were placed on an orbital shaker at 150 rpm for different time periods (20–120 min) at room temperature ~25% ± 2 °C. The equilibrium was achieved at 120 min. The concentrations of Cd^2+^ ions in the filtrate were further diluted and investigated with a graphite furnace atomic absorption spectrophotometer (Model No. GFS97 Thermo Scientific Inc., Borken, Germany).

The adsorption capacities at equilibrium and time t, and percentage removal of Cd^2+^ were calculated as follows:(1)$$Q_{e}\;=\;\frac{\left(C_{i}-{\;C}_{e}\right)\times V}{M}$$
(2)$$Q_{t}\;=\frac{\left(C_{i\;}-{\;C}_{t}\right)\times V}{\;M}$$
(3)$$R\%=\;\left(\frac{C_{i}-{\;C}_{t}}{C_{i}}\right)\times 100$$
where Q_e_ and Q_t_ are the quantities of metal ions (mg/g) eliminated at equilibrium and at time t, C_i_ (mg L^−1^) is the initial concentrations, C_e_ (mg L^−1^) is the equilibrium concentrations and C_t_ (mg L^−1^) concentrations at time t of Cd^2+^, respectively, V (L) is the volume of Cd^2+^ solution, M is the adsorbent mass (g), and R is the percentage (%) removal. 

#### Effect of Initial Cd^2+^ Concentration and Time

Metal removal (%) by CS-KAC-Ag was determined at different initial concentrations of Cd^2+^ (10, 20, and 30 mg L^−1^)mg/L) at constant pH 6, with 20 mg of adsorbent at room temperature. The aqueous solutions were shaken at 150 rpm and filtered with a Whatman filter paper (3 µm).

### 3.2. Reusability Study of CS-KAC-Ag 

The recycling study was performed to explore the economic feasibility of the nano-bioadsorbent.

The reusability studies of the CS-KAC-Ag adsorbent were investigated using the adsorption/desorption cycles. This was conducted using the optimum experimental parameters (pH~6.5, adsorbent dosage 20 mg, time of contact 120 min) attained during optimization. Desorption of the analytes from the adsorbent was achieved by washing the adsorbent with 2 M nitric acid. In addition, deionized water was used to wash the adsorbent before drying it in an oven and reused. The cycle of the adsorption–desorption process was repeated until the adsorbent was exhausted.

### 3.3. Adsorption Kinetics of Cd^2+^


To evaluate the kinetics behavior in the adsorption process of Cd^2+^ by CS-KAC-Ag, pseudo-first-order kinetics and pseudo-second-order kinetics models were utilized and fitted to the experimental data [23].
(4)$$\ln\;(Q_{e}-Q_{t})=\ln\;Q_{e}-k_{1}t\;$$
(5)$$\frac{t}{Q_{t}}=\frac{1}{k_{2}Q_{e}^{2}}+\frac{t}{Q_{e}}$$
where k_1_ is the rate constant of the pseudo-first-order sorption (min^−1^), k_2_ is the pseudo-second-order sorption (g mg^−1^ min^−1^), and t (min) is the contact time.

### 3.4. Adsorption of Water Samples from Mining Sites in Mexico 

A water sample was collected from the Presa de Jalpan mining sites in Mexico and the efficacy of the nanobiocomposite CS-KAC-Ag was evaluated. The wastewater sample was first filtered using a Whatman 44 filter paper of 3 μm pore size to remove suspended particulate matter. The pH was recorded using a pH meter (Sper Scientific Basic 840087, SPER SCIENTIFIC, Scottsdale, AZ, USA). Calgon, a common CAC was used as a control. The 100 mL of the mining water sample was treated separately with 20 mg each of KAC, CAC, and CS-KAC-Ag for a contact time of 120 min at room temperature. After filtration, each filtrate was tested with an atomic absorption spectrophotometer (Model No. Perkin Elmer A Analyst 300, Thermo Fisher Scientific Inc., Borken, Germany). All other experimental conditions were maintained to compare the adsorption efficiency of the three materials.

### 3.5. Design of Experiments

This study adopted the Box–Behnken design (BBD) of experiments, a widely used response surface method (RSM) in contemporary scientific research [24,25]. This randomized design method was preferred because of its operational factor optimization capacity (variation of factor treatments to determine the ones that would yield the best outcomes), it requires fewer experimental trials, or runs, and its powerful statistical interpretation capabilities. It allows for rigorous and simultaneous testing of various aspects of delivering versatile interventions (treatments) and can also produce higher order of response surfaces. The method allows fitting a second-order polynomial regression equation of the form:(6)$$Y=\beta_{0}+\sum_{i=1}^{n}\beta_{i}X_{i}+\sum_{i=1}^{n}\beta_{\mathrm{ii}}X_{i}^{2}+\sum_{\begin{array}{c} i,j=1\\ i<j \end{array}}^{n}\beta_{\mathrm{ij}}X_{i}X_{j}+\epsilon$$
where Y = response (i.e., dependent variable), $X_{i}$ = factors (i.e., independent variables), $X_{i}X_{j}$ = interaction terms, $\beta_{0}$= intercept, $\beta_{i},{\;\beta }_{\mathrm{ii}},{\;\beta }_{\mathrm{ij}}$= regression coefficients (i.e., main and interaction effects), n = number of factors, and ε = random (error) term.

In this study, the % removal (Y) measured as a percentage represents the response variable, and the 3 factors considered are initial metal ion concentration ($X_{1}$) measured in mg L^−1^, pH $\left(X_{2}\right)$, and contact time ($X_{3}$) measured in minutes. Each factor was coded on 3 levels (−1, 0, 1) according to the formula:(7)$$X_{i}=\frac{x_{i}-x_{0}}{R/2}\;$$
where X_i_ = coded value of the ith factor, x_i_ = actual value of the ith factor, x_0_ = center point value of the ith factor, and R = range of the ith factor values. The resulting BBD code mapping to the actual level is depicted in Table 1. The BBD design has 15 total runs, including 3 center point runs (Table 2). 

#### 3.5.1. Parameters Optimization Using Hybrid RSM-DF Method

An interesting practical insight was to determine the optimal combination of factors X_1_, X_2,_ and X_3_ that maximizes the removal efficiency (i.e., the response variable). Therefore, a response surface optimization analysis was performed, and the results were interpreted using individual and composite desirability scores. It was desired to explore how the factors maximize a single response, while composite desirability (D) was used to optimize multiple responses overall. The desirability scores ranged from 0 to 1, where 1 depicts the ideal case and 0 denotes that one or more responses are outside their acceptable limits. Since this study was to explore a single response variable Y (i.e., percentage removal), it was expected that the individual (d) and combined (D) desirability scores would be the same. 

#### 3.5.2. Statistical Analysis

The statistical data analysis was performed in Minitab version 19 (developed by Minitab LLC, State College, PA, USA) which has superior computational automation and graphical capabilities. Analysis of variance (ANOVA) was used to determine the significance of linear, squared, and interaction effect terms in the BBD regression model. In this study, an α of 5% or below is considered statistically significant. This corresponds to a confidence level (CL) of 95% or above (1 − α). In other words, *p*-values smaller than 0.05 are considered statistically significant at a CL of 95% or above.

## 4. Results and Discussion

### 4.1. Characterization of CS-KAC-Ag

#### 4.1.1. Ultraviolet-Visible Spectroscopy

The UV-Vis spectrum (wavelength ranging from 200 to 800 nm) of the synthesized nanobiocomposite is shown in Figure 2. The formation of AgNPs was identified by the visual color change (inset dark brown A) and the surface plasmon resonance peak was observed at around 424 nm within 40 min of photoirradiation indicating the formation of AgNPs within the CS–KAC matrix. 

#### 4.1.2. BET Analysis

The Brunauer–Emmett–Teller (BET) specific surface areas of KF, KAC, and CS-KAC-Ag were determined and are summarized in Table 3. The data showed the BET surface area and pore volume of CS-KAC-Ag were less than that of KAC. This result could be due to the successful loading of AgNPs on the surface of KAC. BET pores are usually categorized into three main categories (International Union of Pure and Applied Chemistry, 1972) depending on pure diameters; pore size < 2 Å are designated as micropores, pore size 2–50 Å are mesopores, while pore size > 50 Å are macropores. The fabricated novel adsorbent exhibited a pore diameter of size 3.2 nm and hence could be categorized as mesoporous in nature. 

#### 4.1.3. SEM Analysis

The surface morphologies of KF, KAC, and CS-KAC-Ag were investigated using SEM analysis, and the results are shown in Figure 3. The surface of KF shown in Figure 3a was dense, uneven, and rough. As shown in Figure 3b, after the activation, multiple pores were exhibited on the KAC surface as expected. As observed, compared with the KAC, the surface morphology of CS-KAC-Ag was substantially different with fewer pores. This suggested that the surface pores were blocked by AgNPs or silver ions during the growth of AgNPs within the KAC matrix. Some SEM images of CS-KAC-Ag exhibited spherical shape nano/microstructures hypothesized to be AgNPs that formed in situ within the KAC matrix. From the SEM images from Figure 3c,d (×5000), the average particle size of the nano-biocomposite was calculated at ~150 nm. The larger size of the AgNPs would be due to the aggregation of AgNPs within the nanocomposite matrix and also the heterogeneity of the stabilizing matrix. 

The EDS mapping of the CS-KAC-Ag synthesized at room temperature is presented in Figure 3e. It was observed from Figure 3e that a typical absorption peak was observed approximately at 3 keV for AgNPs due to the surface plasmon resonance, which demonstrated the presence of silver metal.

#### 4.1.4. FTIR Analysis

The functional group of KAC and the CS-KAC-Ag nano-biocomposite can be better understood from the FTIR study. The FTIR spectra of the KAC and CS-KAC-Ag are shown in Figure 4. For kenaf-based activated (KAC), the adsorption band around 3671 can be attributed to the single bond OH stretch vibration confirming the presence of hydroxyl groups in the sample [26]. The adsorption band noted around 2979 and 2902 cm^−1^ confirmed the stretching vibrations of C-H due to the presence of cellulose and hemicellulose [26]. The peaks of KAC at 1401 and 1244 cm^−1^ were due to the C=O stretching vibration for aldehydes and carboxyl groups [27]. The characteristic peak of KAC at 1062 cm^−1^ was traced to the small peaks for KAC at 884 cm^−1^ that was due to the presence of C=C (alkene) and halo compound [27]. Peaks at 3413, 3264, 1630, 1337, 1149, 1023, and 825 cm^−1^ were observed in the spectrum of the nanocomposite CS-KAC-Ag. The vibration peaks at 3413 and 3264 cm^−1^ after the formation of the nano-biocomposite as shown in Figure 4 were assigned due to the O–H and N–H stretching vibration and amine groups [28]. Amino (NH_2_) group of chitosan where CS acted as a stabilizing medium and modified KAC with CS introduced a lot of amine and hydroxyl functional groups. A sharp small peak of CS-KAC-Ag at 1630 cm^−1^ was due to the vibration of the N–H bond groups [29]. A strong peak at 1337 cm^−1^ was ascribed to the C–N stretching of aromatic amine [30]. In the FTIR spectra of CS-KAC-Ag, the absorption band noted at 1149 cm^−1^ was ascribed to the C–O stretching vibrations of alcohols [30]. Another characteristic peak observed at 1023 cm^−1^ may be attributed to the C–N stretching amine or the presence of polysaccharides [31]. The FTIR spectra of CS-KAC-Ag contained new peaks at 825 cm^−1^ and 600 cm^−1^, and these adsorption peaks may be attributed to the chemical interaction and the formation of metal–oxygen bonds between the functional groups of KAC, biopolymer chitosan, and silver ions during the fabrication of the nano-biocomposite [32]. 

### 4.2. Statistical Results and Interpretation

ANOVA and regression analysis were performed in Minitab v. 19 and the results are presented in Table 4. The predictive second-order polynomial model was a good fit for the data and was statistically significant (F = 20.92, *p*-value = 0.002) at α = 5% (CL = 95%).

Most individual main effects and interaction terms were also significant except for X_1_ × X_1_ (F = 1.62, *p*-value = 0.259) and X_1_ × X_2_ (F = 0.02, *p*-value = 0.895), as shown in Table 4. 

The explanatory power of the model was high (R-sq = 0.97), indicating that 97.4% of the response variability was explained by the combination of the variabilities of the factors X_1_, X_2_, and X_3_. The model was further evaluated for multicollinearity among the variables using the variance inflation factors (VIF) shown in Table 4. The resulting VIF factors were less than 5, indicating that multicollinearity was not a concern [33,34]. The resulting regression expression is provided in Equation (8) and is used to predict the response variable Y_pred_ in Table 5 considering the coded variables to preserve the orthogonality of the design. The experiment has been run according to the BBD design and the experimental results Y_exp_ are provided in Table 5.
Y = 85.17 − 15.00 X_1_ + 5.68 X_2_ + 13.93 X_3_ − 3.5 X_1_ × X_1_ − 13.16 X_2_ × X_2_ − 8.13 X_3_ × X_3_ − 0.37 X_1_ × X_2_ + 11.94 X_1_ × X_3_ + 7.86 X_2_ × X_3_(8)

The main factors in cadmium removal are shown in Figure 5a. As the individual factors are varied, there are significant main effects, and the sharper the slope of the line, the more important the main effect. The plot suggested that all first-order factors significantly impacted the average response Y. The interaction effects are shown in Figure 5b. The parallel lines indicated no interaction. The larger the slope difference (in absolute value) between the lines, the higher the magnitude of the interaction. The plot suggested significant 2-way interactions except for that between X_1_ and X_2_, which was consistent with the ANOVA results shown in Table 4. A Pareto chart of the standardized effects was created to rank the effects by their magnitude, as illustrated in Figure 5c. If a bar passes the vertical red dotted line at the average value of 2.571, the corresponding effect is significant, and the larger the bar, the more important that effect is. The plot showed that the X_1_ × X_1_ and X_1_ × X_2_ effects were not significant (the standardized effect size is less than the 2.571 threshold), and the effects impacted on the response variable Y decrease in the following order: X_1_, X_3_, X_2_ × X_2_, X_1_ × X_3_, X_2_, X_2_ × X_3_, X_3_ × X_3_. These results were consistent with the ANOVA results in Table 4, where the *p*-values for X_1_ × X_1_ and X_1_ × X_2_ are both greater than 0.05 (the statistical significance), and the results in Figure 5b, where the plotted lines for X_1_ × X_2_ are almost parallel (indicating a non-significant interaction). This provides empirical confirmation that X_1_ (initial metal concentration), X_2_ (pH), and X_3_ (contact time) are significant factors influencing Y (% removal) and that there are certain significant interactions at play as well. For example, Table 4 shows a significant interaction between X_2_ and X_3_ (*p*-value less than 0.05), and Figure 5b also confirms it (non-parallel lines on the X_2_ × X_3_ plot). Moreover, Figure 5b gives a sense of the nature of that interaction: when X_1_ (initial metal concentration) stays at 0 (or 30 mg L^−1^), for the same level of X_2_ (pH), the levels of Y would vary differently as X_3_ (contact time) varies. For example, when X_1_ (initial metal concentration) = 0, or 30 mg L^−1^, for the same X_2_ (pH) = 1, or a pH of 9, the Y (% removal) increases as X_3_ (contact time) increases (from −1 to 0 to 1, or from 20 to 70 to 120 min, respectively). However, when X_2_ = −1, or a pH of 3, Y decreases when X_3_ increases from 0 (70 min) to 1 (120 min). These results are also confirmed by Table 5 when using (in X_1_/X_2_/X_3_ format) 30/6/70 as a proxy for the missing combinations (by BBD design) 30/9/70 and 30/3/70. A similar interpretation is also found for the X_1_ × X_3_ interaction, based on the results in Table 5 and Figure 5b. These findings lead one on the path of finding an optimal combination of the three parameters X_1_, X_2,_ and X_3_ that would maximize the % removal. This discussion is addressed in the surface and contour plots analysis that follows, and in Section 4.3 and Section 4.4 of this study. 

The surface and contour plots shown in Figure 6a–f demonstrate the impact of two factors while keeping the third factor constant at the center point. The plots revealed that the removal of Cd^2+^ significantly increased as the initial metal concentration X_1_ was lowered around the minimum of 10 mg L^−1^. The solution alkalinity X_2_ was increased beyond a pH of 4.5 while keeping the contact time X_3_ at around 70 min. Similarly, the removal increased to more than 90% when the initial metal concentration X_1_ was lowered towards the minimum of 10 mg L^−1^, and the contact time X_3_ was increased from 20 min to 33 min when the pH of the solution was maintained at 6. No significant improvement in the removal of the metal ions was observed with the increase in contact time. When the initial metal ions (Cd^2+^) concentration was kept constant at around 30 mg L^−1^, the % removal of Cd^2+^ escalated remarkably when the contact time X_3_ increased from 70 to 83 min while the pH of the solution X_2_ was kept at a pH of 6 and above. Interestingly, as long as the contact time was above 83 min, the pH increases beyond 6 would not make a major difference in the removal, which remained relatively flat at around 90%.

In conclusion, the key findings are the statistical significance of the X_1_, X_2_, and X_3_ factors (i.e., main effects), and of the combination of factors X_1_ × X_3_ and X_2_ × X_3_ (i.e., interactions), which together influence the % removal. 

### 4.3. Interaction Effect on the Removal of Cd^2+^

The contour plots and the response surface are shown in Figure 6, which explain the relationship between the variables and the efficiency of the adsorbent in the removal of Cd^2+^. The effect of metal concentration and pH on the decontamination of Cd^2+^ is shown in Figure 6a. The pH is an important parameter, and its values affect the adsorbent’s surface charge, at times degree of ionization of the adsorbate during adsorption. Thus, the effect of pH (H^+^ concentration) in the solutions on the adsorption % of Cd^2+^ was studied at different pH values and it was observed that the Cd^2+^ removal increased with the increase in pH of the solution and decreased with the increase in metal ion concentration. Similar results were obtained for Cd^2+^ adsorption on zinc oxide activated carbon nanocomposite at pH 9, and the removal was 95.1% for 10 mg L^−1^ of Cd^2+^ concentration. In an acidic medium, with a low pH (~3), the availability of H^+^ ions increased, which led to a repulsion between Cd^2+^ and active sites on the surface of the nanocomposite as reported by other researchers [35]. Therefore, a decline in Cd^2+^ adsorption was observed at a low pH. The higher the pH value, the more the OH ions, thereby forming a negatively charged surface on the CS-KAC-Ag to attract the positively charged Cd^2+^. The removal capability of CS-KAC-Ag decreased with the increase in Cd^2+^ concentrations initially due to higher available adsorption sites on CS-KAC-Ag. Similarly, Figure 6b,c show the effect of the initial metal concentration and the contact time, and the effect of pH and exposure time, respectively, on the % removal. Figure 6d–f are the corresponding contour plots.

The study of these main effects and interactions suggests that an optimal combination of factors does exist that maximizes the % removal and that this combination can be calculated as shown in the next section. 

### 4.4. Optimization Results and Interpretation 

A process optimization under the hybrid RSM–DF method was carried out and the results are shown in Figure 7. The optimum coded values for the initial metal concentration X_1_, solution alkalinity X_2,_ and contact time X_3_ were −1, 0.3131, and 0.2727, respectively, corresponding to the initial metal concentration of 10 mg L^−1^, solution pH of 6.9393, and a contact time of 83.635 min, which delivered a removal of 97.9%. As expected, the individual desirability (d) obtained under these conditions was equal to 1, while the composite desirability (D) was 0.9691, which is close to 1, thus confirming the reliability and robustness of the optimal parameters.

### 4.5. Adsorption Kinetics

Experiments were carried out to explore the effect of contact time on the adsorption efficiency of CS-KAC-Ag. It was observed from Figure 8 that the removal (~70%) increased rapidly within the first 40 min for 10 mg L^−1^, 30 mg L^−1^, and 50 mg L^−1^ of Cd^2+^ concentrations which aligns very well with the statistical analyses. Within 120 min of contact time, the adsorption reached the equilibrium state. The removal of Cd^2+^ was 91.8%, 90.4%, and 84.1% for 10 mg L^−1^, 30 mg L^−1^, and 50 mg L^−1^ of metal concentrations, respectively, at pH~6.5 and using 20 mg of the nano-biocomposite. A large number of reactive atoms and more vacant reactive metallic spots correspond to a higher adsorption rate initially, which gradually diminished with a declination of vacant sites with an increase in contact time.

The pseudo-first-order rate constant k_1,_ pseudo-second-order rate constant k_2_, and the linear correlation coefficient R^2^ are shown in Table 6.

Table 6 shows that the R^2^ values of pseudo-first-order and pseudo-second-order models were 0.850 and 0.998, respectively. The higher R^2^ value for the pseudo-second-order for Cd^2+^ indicated that the kinetic adsorption fitted well with the pseudo-second-order model. The adequacy of the pseudo-second-order model to explain the adsorption kinetic data was also established by comparing q_e,_ (exp: 47.12 mg/g) and q_e_ (cal: 46.95 mg/g) values.

### 4.6. Application to Mining Wastewater Sample

Wastewater from mining was used as the treatment target. A wastewater sample was collected from Presa de Jalpan mining sites in Mexico. Although Cd (II) was not detected in the mining water, presence of similar metallic pollutants such as Cobalt Co (II), Chromium Cr (VI), and Nickel Ni (II) was detected in the water sample and these metallic contaminants were utilized to confirm the superior efficiency of the developed novel adsorbent. Figure 9 shows the removal % of the three metals using KAC, CAC, and nano-biocomposite CS-KAC-Ag. The removal efficiency of Cr (VI) 91.7% > Ni (II) 84.4% > Co (II) 80.5% was observed with the nano-biocomposite (Figure 9). It is reported that the ionic radius of Cr (VI) (0.44 Å), Ni (II) (0.56 Å), and Co (II) (0.56 Å), respectively, and hence metals with smaller ionic radius diffused faster [35]. Thus, it is observed that metal ions with lower ionic radius diffuse faster onto the binding sites of the fabricated nano-bioadsorbent, compared with those metals with higher ionic radii. Though CAC exhibited similar performance towards the removal of metals ions using mining wastewater, it is not an environmentally friendly adsorbent and hence the sustainable synthesized nano-biocomposite proved to be more effective even at lower dosage (0.2 g L^−1^) to minimise the production of toxic waste and remove multiple toxic metals ions including higher concentrations Cd^2+^ ions. 

### 4.7. Reusability Test of the Nano-adsorbent

A reusable adsorbent can minimize the operational cost in real-life application; hence, the reusability study of the synthesized adsorbent is explored. Using a Cd^2+^ concentration of 50 mg L^−1^, and pH~6.5, the recyclability test of the nano-adsorbent was performed. The removal for cycle 1 was ~87.9%, while for cycle 4 ~87.0%. The results indicated that the adsorbent preserved the same metal adsorption capacity even after four cycles of reuse (Figure 10). The maximum variation in removal percentage of the first and fourth adsorption cycle was found to be less than 3%, suggesting that the porous CS-KAC-Ag systems had a stable efficacy. This aspect is particularly important because of the current stringent ecological and economic regulations for sustainability and safety. 

### 4.8. Adsorption Mechanism 

The adsorption mechanism of Cd^2+^ and the other three metals Cr^6+^, Co^2+^, and Ni^2+^ onto the CS-KAC-Ag surface can be attributed to different types of interactions among different functional groups, such as ion exchange, electrostatic interaction, or interaction between oxygen surface groups and metal ions by the formation of hydrogen bonding [36]. The amino groups, one of the most active functional groups of CS, may be a contributing factor, and protonation of amine groups often leads to an exchange of ions and electrostatic attraction between neighboring ions. The presence of free electrons on the nitrogen of the amino group might be responsible for the uptake of metal ions by the chelation mechanism [37]. The –OH and –NH_2_ functional groups in the nano-biocomposite also contributed to the reduction of metal ions. Moreover, it is reported that silver readily gets adsorbed on CS and CS acts as a crosslinker to develop hybrid chitosan-based nanocomposite. When AgNPS were dispersed on the activated carbon surface, the synthesized material’s surface area increased in comparison to bare nanoparticles. However, the overall surface area of the nano-biocomposite as reported in the BET surface area of the nano-biocomposite in comparison to KAC has diminished due to the impregnation of AgNPs. The nano-biocomposite (CS-KAC-Ag) formed with AgNPs loaded on CS and KAC-based matrix with high porosity and multiple functional groups exhibited a superior adsorption capability towards heavy metal ions. Silver nanoparticles immobilized on CS and KAC increased the number of active sites or electron-rich sites such as C-O, C-N, N-H, C=O, C=C, or O-H as reported by the FTIR study on the nano-biocomposite surfaces due to which superior adsorption capability was observed towards Cd^2+^ in comparison to KAC as shown in Figure 11. 

### 4.9. Comparison with Other Nanobiocomposites

A comparison of our fabricated adsorbent with other studied adsorbent materials for the removal of Cd^2+^ from wastewater is presented in Table 7. It was observed that the adsorption efficiency of our synthesized material was high (95.1%), with 0.2 g/L of adsorption dosage, a contact time of 120 min, and a pH of 6.5, which easily mimicked the pH of the real wastewater sample.

## 5. Conclusions

CS-KAC-Ag nano-biocomposite was successfully developed by a facile eco-friendly method using kenaf-based activated carbon. The Cd^2+^ removal efficiency using the nanobiocomposite was investigated. The BET value of the nano-biocomposite was obtained at 204 m^2^/g. The synthesized material exhibited high adsorption efficiency for the purification of mining wastewater containing Co^2+^, Cr^6+^, and Ni^2+^. The scanning electron microscope and the energy-dispersive X-ray results of the adsorbent showed the presence of Ag on the nanocomposite surface. The Box–Behnken design was applied effectively to evaluate the Cd^2+^ ions concentration, the pH, and the contact time. The best optimum parameters for pH, contact time, and adsorbent dose were 9.0, 120 min, and 20 mg, respectively. As per the R^2^ > 0.99 value, the nano-adsorbent exhibited a superior accord associated with the pseudo-second-order model. This novel, cost-effective, sustainable nano-adsorbent could remove 95.1% of Cd^2+^ at the optimum condition and Cr^6+^ 91.7% > Ni^2+^ 84.4% > Co^2+^ 80.5% at pH 6.5, 120 min of contact time, and 0.02 g/L of adsorbent dosage. Moreover, the metal elimination ability of the porous adsorbent remained unaltered over four successive adsorption–desorption cycles. The developed CS-KAC-Ag is sustainable, inexpensive, and reusable material for the remediation of toxic metal ions from wastewater. Furthermore, in future research, this novel adsorbent can be considered for removing organic and microbial pollutants from wastewater.

## Figures and Tables

**Figure 1 materials-15-08901-f001:**
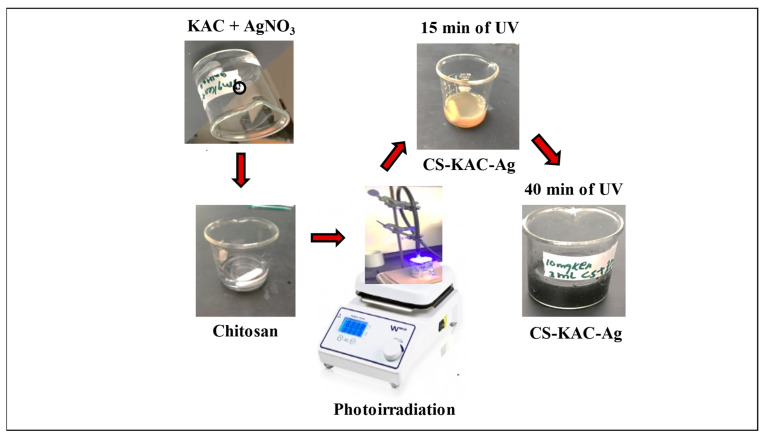
Schematic representation of the formation of silver nanoparticles within chitosan and kenaf-based activated carbon.

**Figure 2 materials-15-08901-f002:**
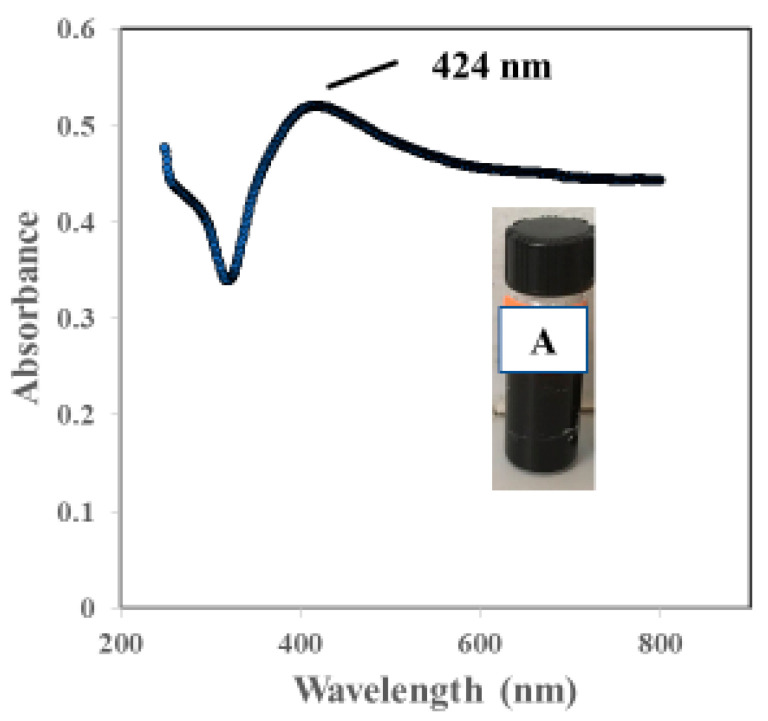
UV-Vis spectrum of the AgNps synthesized by novel method.

**Figure 3 materials-15-08901-f003:**
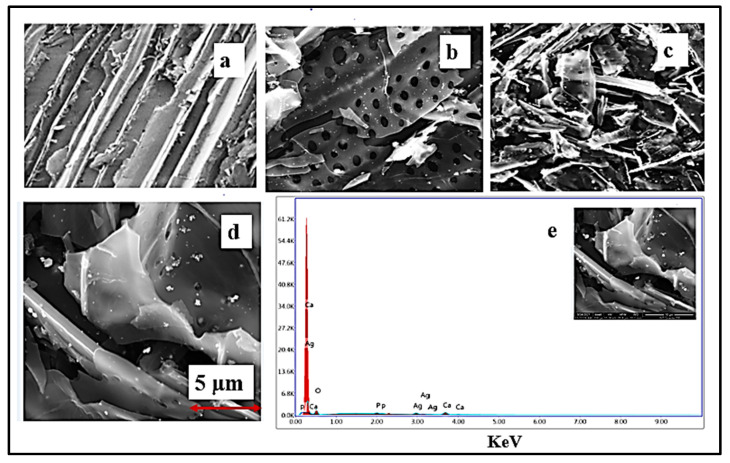
SEM images of (**a**) raw KF × 2000, (**b**) KAC × 2000, (**c**) CS-KAC-Ag × 2000, (**d**) CS-KAC-Ag × 5000, and (**e**) EDX spectra of CS-KAC-Ag.

**Figure 4 materials-15-08901-f004:**
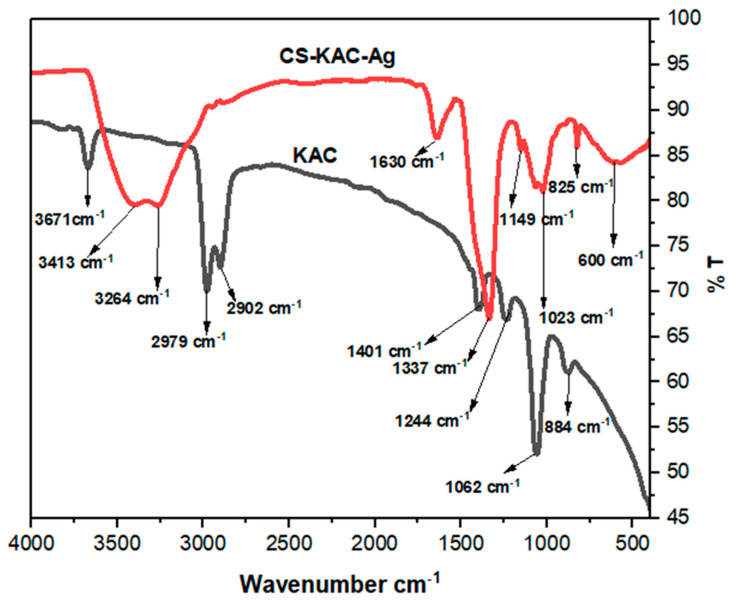
FTIR spectra of KAC and CS-KAC-Ag.

**Figure 5 materials-15-08901-f005:**
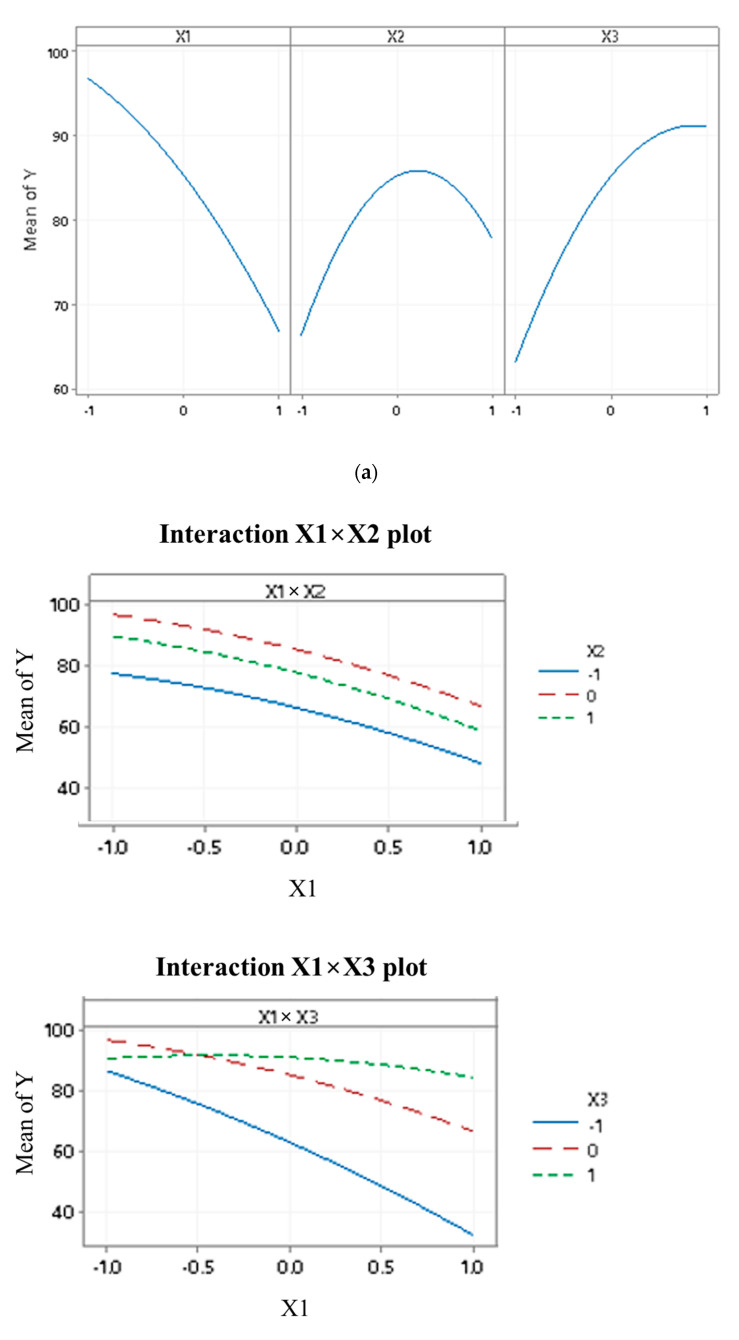
(**a**) Contribution of the main factors X_1_, X_2_, and X_3_ on explaining Y. (**b**) Interaction plots. (**c**) Pareto Chart of parameter effects on the removal efficiency for Cd^2+^.

**Figure 6 materials-15-08901-f006:**
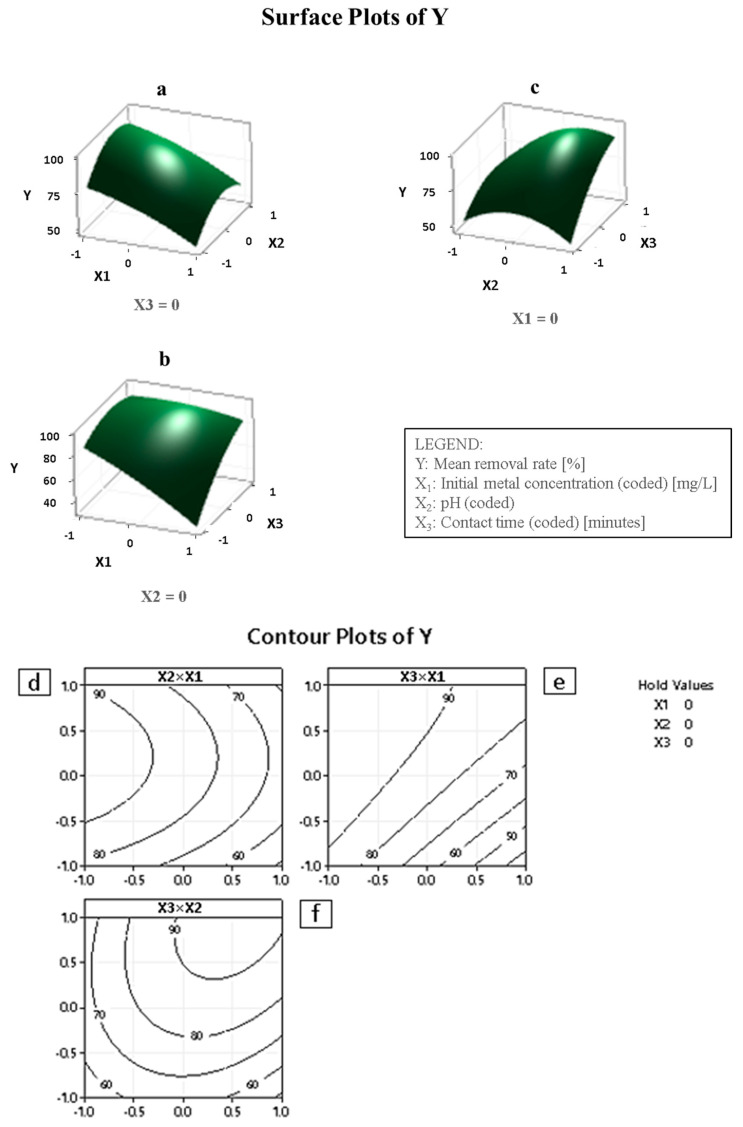
Response surfaces and contour plots for the removal of Cd^2+^ using CS-KAC-Ag. (**a**,**d**) Effects of Cd^2+^ concentration and pH on % removal; (**b**,**e**) Effects of Cd^2+^ concentration and contact time on % removal; (**c**,**f**) Effects of pH and contact time on % removal.

**Figure 7 materials-15-08901-f007:**
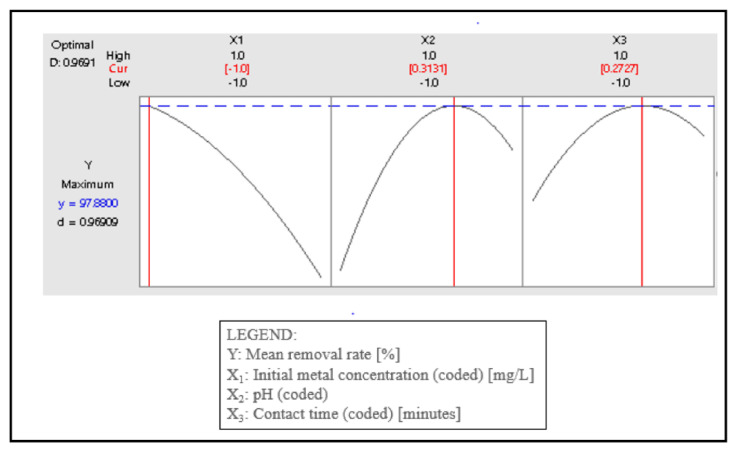
Response optimization results in the removal of cadmium with different contact times, pH, and initial metal concentrations.

**Figure 8 materials-15-08901-f008:**
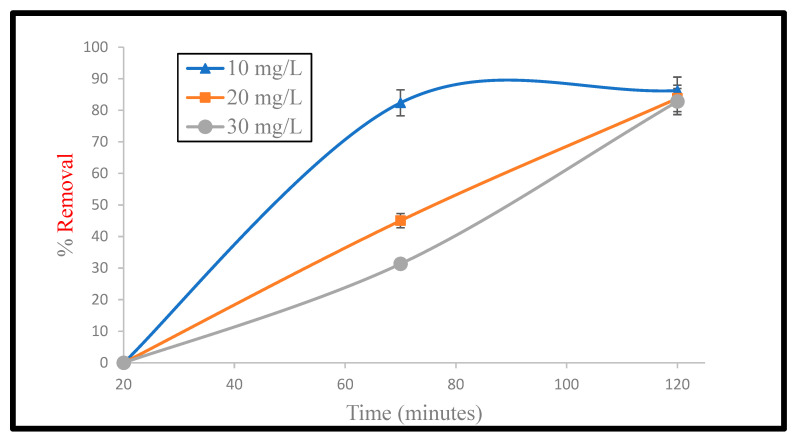
Removal of cadmium at different concentrations (10 mg L^−1^, 20 mg L^−1^, and 30 mg L^−1^) with the change in contact time (20–120 min at 20 mg adsorbent dosage).

**Figure 9 materials-15-08901-f009:**
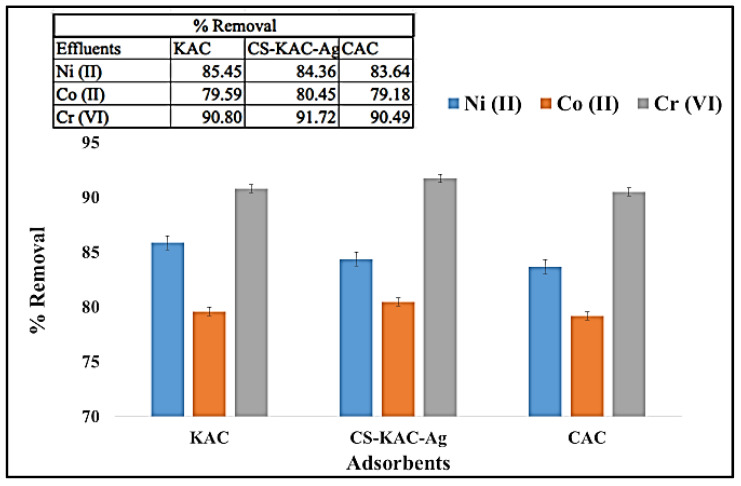
Removal % of Ni (II), Co (II), and Cr (VI) using (KAC), commercial activated carbon Calgon (CAC), and CS-KAC-Ag.

**Figure 10 materials-15-08901-f010:**
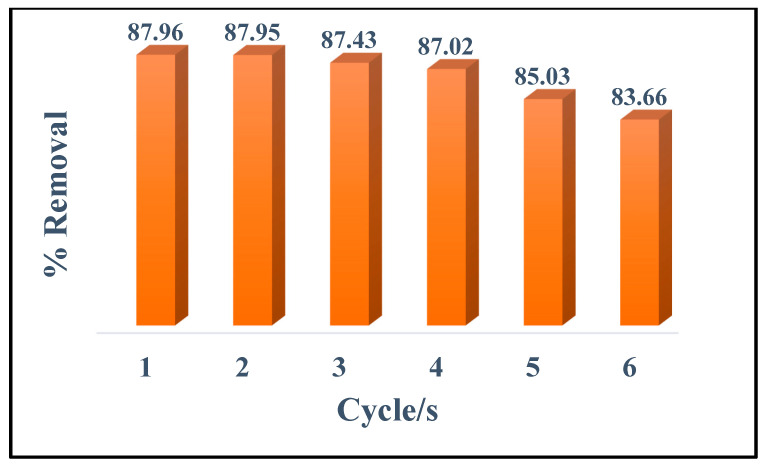
Recyclability test of the nano-adsorbent for the remediation of Cd^2+^.

**Figure 11 materials-15-08901-f011:**
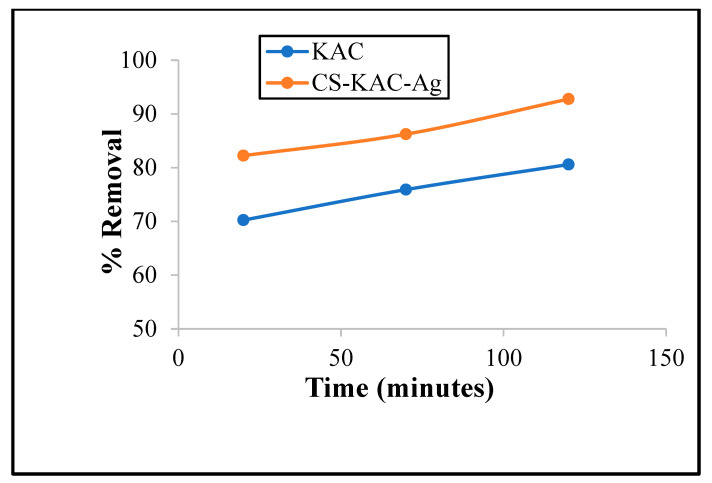
Comparison of % removal of Cd^2+^ using KAC and CS-KAC-Ag (10 mg L^−1^ L of Cd^2+^, 20 g of adsorbent, pH~6, room temperature).

**Table 1 materials-15-08901-t001:** Actual and coded factor levels.

Factors	Level 1		Level 2		Level 3	
	Actual	Coded	Actual	Coded	Actual	Coded
Initial metal concentrations (X_1_)	10 mg L^−1^	−1	30 mg L^−1^	0	50 mg L^−1^	1
pH (X_2_)	3	−1	6	0	9	1
Contact time (X_3_)	20 min	−1	70 min	0	120 min	1

**Table 2 materials-15-08901-t002:** The Box–Behnken (BBD) design of experiments.

Run No.	Coded Values		
	X_1_	X_2_	X_3_
1	1	0	1
2	1	−1	0
3	0	−1	1
4	0	0	0
5	0	−1	−1
6	1	1	0
7	0	1	1
8	0	0	0
9	−1	0	−1
10	1	0	−1
11	0	0	0
12	−1	−1	0
13	−1	1	0
14	−1	0	1
15	0	1	−1

**Table 3 materials-15-08901-t003:** Physio-chemical characteristics of KF, KAC, and CS-KAC-Ag for the adsorption of Cd^2+^.

Physio-Chemical Characteristics	KF	KAC	CS-KAC-Ag
BET specific surface area, m^2^/g	3.2	843.3	204.400
T plot micropore volume, cm^3^/g	0.0	0.157	0.003
T plot micropore area, m^2^/g	0.0	317.9	8.300
Cumulative adsorption surface area (BJH method), m^2^/g	0.0	387.9	151.300
Pore diameter, nm	5.5	3.5	3.200

**Table 4 materials-15-08901-t004:** ANOVA table for cadmium removal with CS-KAC-Ag.

Source	DF	Adj SS	Adj MS	F-Value	*p*-Value	VIF
Model	9	5270.20	585.58	20.92	0.002	
Linear	3	3610.29	1203.43	43.00	0.001	
X_1_ (initial metal concentration)	1	1800.00	1800.00	64.32	0.000	1
X_2_ (pH)	1	258.21	258.21	9.23	0.029	1
X_3_ (contact time)	1	1552.08	1552.08	55.46	0.001	1
Square	3	842.55	280.85	10.04	0.015	
X_1_ × X_1_	1	45.30	45.30	1.62	0.259	1.01
X_2_ × X_2_	1	639.45	639.45	22.85	0.005	1.01
X_3_ × X_3_	1	244.35	244.35	8.73	0.032	1.01
2-way interactions	3	817.36	272.45	9.74	0.016	
X_1_ × X_2_	1	0.54	0.54	0.02	0.895	1
X_1_ × X_3_	1	570.02	570.02	20.37	0.006	1
X_2_ × X_3_	1	246.80	246.80	8.82	0.031	1
Error	5	139.93	27.99			
Lack-of-fit	3	139.93	46.64			
Pure Error	2	5270.20	585.58	20.92	0.002	
Total	14	3610.29	1203.43	43.00	0.001	

**Table 5 materials-15-08901-t005:** Predicted and experimental results.

Run No.	Coded Values			Actual Values			Predicted % Removal	% Removal
	X_1_	X_2_	X_3_	X_1_ [mg L^−1^]	X_2_	X_3_ [min.]	Y_pred_ [%]	Y_exp_ [%]
1	1	0	1	50	6	120	84.41	84.08
2	1	−1	0	50	3	70	48.2	47.12
3	0	−1	1	30	3	120	64.27	62.83
4	0	0	0	30	6	70	85.17	83.77
5	0	−1	−1	30	3	20	52.13	51.31
6	1	1	0	50	9	70	58.82	54.45
7	0	1	1	30	9	120	91.35	91.10
8	0	0	0	30	6	70	85.17	83.77
9	−1	0	−1	10	6	20	86.55	82.36
10	1	0	−1	50	6	20	32.67	31.41
11	0	0	0	30	6	70	85.17	83.77
12	−1	−1	0	10	3	70	77.46	77.43
13	−1	1	0	10	9	70	89.56	87.49
14	−1	0	1	10	6	120	90.53	89.89
15	0	1	−1	30	9	20	47.77	46.07

**Table 6 materials-15-08901-t006:** Constants of kinetic models for Cd^2+^ adsorption at pH 6, temperature 25 ± 2 °C, and contact time 120 min.

Kinetic Models	Parameters	Cd^2+^
Pseudo-first-order	q_e_ (mg g^−1^)	16.234
	k_1_ (min^−1^)	0.037
	R^2^	0.850
Pseudo-second-order	q_e_ (mg g^−1^)	46.948
	k_2_	0.021
	R^2^	0.998

**Table 7 materials-15-08901-t007:** Operating values and Cd^2+^ removal percentage using nano-biocomposites.

No.	Adsorbent	Concentration (mg L^−1^)	pH	Adsorbent Dosage (g/L)	Removal Efficiency %	Literature
1	Chitosan/activated carbon iron nanocomposite	5	4.2	0.03	95	[38]
2	ZnO/activated carbon	100	7	0.10	80	[39]
3	Graphene Oxide−Zirconium Phosphate (GO−Zr-P) Nanocomposite	50	6	0.15	99	[40]
4	Lemon-activated carbon with Fe_3_O_4_	10	6	0.14	98.1	[41]
5	Silica aerogel-activated carbon composite	3	6	0.10	64.2	[42]
6	Iron oxide-modified clay-activated carbon composite	5	4.5	2.00	70.0	[43]
7	Maize tassel-magnetite nanohybrid	45	3.5	5.3	97.5	[44]
8	CS-KAC-Ag	10	6.5	0.20	95.1	This study

## Data Availability

Not applicable.

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
