# Peer review of "Lignocellulosic-Based Activated Carbon-Loaded Silver Nanoparticles and Chitosan for Efficient Removal of Cadmium and Optimization Using Response Surface Methodology"

_materials, 2022, doi:10.3390/ma15248901_

Round 1

Reviewer 1 Report

PDF file was attached.

Reviewer 2 Report

This article reports a study to investigate the performance of a material on cadmium adsorption associated with system optimization. Concrete results were shown from carefully performed experiments and analyses. However, very obvious and important problems are found. Please see the comments shown below. In addition, line numbers should be placed throughout the article.

General comments:

The authors described the results, but did not make adequate discussion. In-depth discussion related to mechanisms is definitely necessary.

When making discussion, please try to compare the results to previous studies and please show references.

Specific comments:

Page 1, Abstract, “The lab data were also corroborated from tests conducted using the actual water samples collected from mining sites in Mexico”: Do you mean that the results shown previously were from artificial water? Please indicate this clearly.

Page 1, the last line, “Several studies suggested that adsorption was one of the most effective…”: You could start a new paragraph from here.

Page 2, “Recently, silver nanopar-ticle (AgNPs) loaded commercial AC matrix … and mixed metals from industrial wastewater [18, 19]”: The logic and coherence of this part should be improved. It reads like you just placed some discrete sentences there.

Page 4, “The wastewater sample was first filtered using a Whatman 44 filter paper to remove suspended particulate matter”: Indicate the pore size of the filter.

Page 4, “…for a contact time of 120 minutes at room temperature”: Why did you use 120 min? Did the adsorption reach equilibrium?

Page 4, “… because of its operational factor optimization capacity”: Show more information of the merit of this method.

Page 5, Table 1: The contents in the same row should be at the same level.

Page 5, Table 2: Table 2 contains results. It may not be good to show results in this section. It would be better to divide the table into two.

Page 5, “The UV–vis spectra of CS-KAC-Ag are shown in Figure 1, the changes in color of the CS-KAC-Ag sample after 15 minutes (Inset Figure 1a) and 40 minutes…”: Why did you show the pictures at 15 and 40 min? Do you show them as examples? Or is there any special reason? Please indicate this. At what time point of the adsorption corresponds to the spectra in Figure 1? Can the colour shown in the insets be analyzed using professional instruments, rather than naked eyes? After showing the results, please make necessary discussion.

Page 6, Table 3: Please show standard deviations of the results.

Page 6, “This suggested that the surface pores were blocked by AgNPs or silver ions during the growth of AgNPs within the KAC matrix”: Is this blockage good or not for adsorption capacity?

Page 7, “The functional group of KAC and CS-KAC-Ag nanobiocomposite can be better understood from the FT-IR study”: Please improve the writing of this paragraph. It is wordy and boring. A table or figure could be good to represent what is discussed here clearly. When discussing the observation, you need to show references.

Page 7, “…would be ascribed to the presence of cellulose and hemicellulose”: Discussion like this, not limited to this discussion, should be expanded. Please explain why and how you drew this conclusion from the observation. Please indicate the references. Please check the whole article according to this comment to make improvements.

Page 7, “…which were attributed to the chemical interaction between the functional groups of AC, chitosan molecules, and silver ions”: Again, discussion like this should be expanded. Indicate how you got this conclusion, any mechanisms, and the references. Please check the whole article for this comment.

Page 8, “The resulting regression expression is provided in Eq. (8) used to predict the response variable Ypred in Table 2 considering the coded variables to preserve the orthogonality of the design”: The model like this is very site-specific. It cannot be used in other situations. Please make discussion on what the model would be like in general.

Page 9, Table 4: It would be better to indicate what X1, X2, and X3 mean in the table to facilitate understanding.

Page 9, “The plot showed that the X1*X1 and X1* X2 effects were not significant…were consistent with the ANOVA results in Table 4”: What mechanisms lead to such results? Please make discussion. You primarily described the results but did not make in-depth discussion, especially that related to mechanisms. In addition, when possible, please make a comparison to other relevant studies.

Page 10, Figure 4: Figure 4 needs a title. When possible, please show statistical analysis results, such as confidence intervals.

Page 10, Figure 4c: Do you really need A, B, and C? X1 to X3 are already quite simple. You need to indicate what X1 to X3 stand for.

Page 11, “The surface and contour plots shown in Figure 5a and Figure 5b demonstrate the impact of two factors while keeping the third factor constant at the center point”: Again, you described the results in this paragraph, but did not make adequate discussion. Please discuss the reasons/mechanisms that could lead to such results.

Page 11, “It was observed that the Cd2+ removal rate increased with the increase in pH of the solution and decreased with the increase in metal ion concentration”: Is this consistent with other studies? Please discuss this.

Page 11, “… which led to a repulsion between Cd2+ and active sites on the surface of the nanocomposite”: For this kind of discussion, you used mechanisms developed in other studies. Please show references.

Page 12, “The optimum coded values for the initial metal concentration…which delivered a removal rate of 97.9 %”: X1 was -1, which means the optimal conditions could be out of the range tested. Please discuss how you respond to this issue.

Page 13, “It was observed from Figure 7 that the adsorption rate increased rapidly with the time”: It seems not right. Please double check this statement.

Page 14, Figure 8: Why did you not show the first-order model in the figure? The figure seems very simple. It can be removed.

Page 14, “Table 5 showed that the R2 values of pseudo-first-order and pseudo-second-order models…”: Please review other studies to discuss whether second-order model is appropriate or not, as indicated by other studies.

Page 14, “Figure 9 shows the removal percentage of the three metals using KAC, CAC, and nano-biocomposites CS-KAC-Ag”: Based on Figure 9, the three absorbents have similar efficiency. So why did you propose this new material?

Page 14, “The removal efficiency of Cr (VI) 91.7 % >Ni (II) 84.4 % > Co(II) 80.5 % was observed with the nano-biocomposites”: Is the results consistent with other studies? Why is the efficiency for Cr higher than Ni and Co? Please discuss the mechanism.

Page 15, Figure 9: Please show error bars to represent standard deviations.

Page 16, Figure 10: Improve this figure. The blue bars are not necessary. A curve is better than the histogram.

Page 16, “It was observed that the adsorption efficiency of our synthesized material was high…”: Please discuss the mechanism why your method is better.

Reviewer 3 Report

Journal: Materials

Title: Lignocellulosic based activated carbon loaded silver nanoparticles and chitosan for efficient removal of cadmium and optimization using Response Surface Methodology

Authors: Sujata Mandal 1, Sreekar B Marpu 2, Mohammad A. Omary 2, Catalin C. Dinulescu 3 Victor Prybutok 4 and Sheldon Q Shi *5

The authors of this manuscript studied the synthesis and characterization of an adsorbent for cadmium removal and optimization. After detailed reviews, the novelty of the study has not been adequately explained by the authors. In addition, it was observed that some parameters examined for adsorption studies were not studied and sufficient optimization data were not obtained. In addition to all these, all the images and graphics provided in the manuscript are inconsistent and visually very poorly presented. In addition to the material method problems and visual deficiencies made within the scope of the whole study, detailed comments and questions are given below. Given all these problems and question marks, I am afraid not recommend publishing the manuscript.

Detailed comments:

The English of the Manuscript is not sufficient. The entire manuscript should be reviewed.

The last sentence of the Introduction should explain the novelty of the work. Written like a brief summary of the material method.

The purity of the chemicals used and obtained should be given.

Preparation of CS-KAC-Ag is given very simply. It can be further elaborated with a schematic.

Adsorption kinetics were studied with 10 mg/L Cd. Is this value a value that can be found in wastewater? If so, it can be specified in the introduction section.

Table 1 is visually incorrect. It should be corrected.

Figure 1. The graphical representation is insufficient. The gaps between 0-200 and 800-1000 nm given for wavelength are very large. Also, there is no caption for the image inside the figure, it is not clear what we should understand.

Figure 2. EDX spectra not readable. Also, what does the SEM image in Fig2.(e) belong to?

The explanation in the text for the EDS image is insufficient.

There is no visual match between the charts provided in the Manuscript. Some graphics are prepared in different fonts and points. I highly recommend revising the graphics.

Fig4 is very simple and not prepared in an acceptable way. It is impossible to comment on interaction plots.

Adsorption kinetics; What is the reason for the trend for 10mg/L? What is the difference between the slope obtained at 60 minutes and the slope obtained later?

Why was the temperature kept constant at 27 degrees? Is there a special reason why 27 degrees was chosen?

The figure is incompatible with the caption of Fig 9. In addition, the results obtained with CAC are almost identical to the results obtained with KAC-Ag. Why should we choose this adsorbent? Its superior features are not understood.

Where are the results of the experimental studies on adsorbent dosage? It is not clear why 20mg was chosen.

One of the most important parameters for adsorption is temperature. Why are there no studies on this parameter?

Reviewer 4 Report

The authors developed a novel silver nanocomposite for Cd ions removal from waterbody. A series of characterizations were applied to investigate the physical and chemical properties for this absorbent, and the response surface methodology was used to optimize the adsorption process, and the kinetics simulation was used to study the adsorption behavior. I consider it could be published after some minor revisions.

1. What was the load capacity of Ag on the composite?

2. The resolution of all figures should be improved. Besides, the font size should be increased in many figures. The EDX in figure 2 should be revised.

3. Did the authors consider the effect of water matrix on Cd ions removal? Such as various anion ions.

4. What was the reason and mechanism for the improvement of CS-KAC-Ag compared to KAC? The discussion should be enriched and deepened.

5. The inset table of Figure 9 should indicate the % removal.

Round 2

Reviewer 2 Report

I appreciate the authors’ great efforts to respond to my comments and improve the article. I have some follow-up comments.

Response to Page 4, “…because of its operational factor optimization capacity”: The information of the merit is quite brief. Please show more details.

Response to Page 5, Table 1: You showed coded and actual values, but there are three columns. It is very confusing. Please improve the table.

Response to Page 5, Table 2: Since you now display predicted and experimental values both in the Results section. It is not good to split them. Please combine them together to become a single table.

Response to Page 5, “The UV–vis spectra of CS-KAC-Ag are shown in Figure 1, the changes in color of the CS-KAC-Ag sample after 15 minutes (Inset Figure 1a) and 40 minutes…”: You have not addressed all comments. Can you show instrumental measurements of colour, rather than the pictures judged by eyes?

Response to Page 8, “The resulting regression expression is provided in Eq. (8) used to predict the response variable Ypred in Table 2 considering the coded variables to preserve the orthogonality of the design”: I may not express my meaning clearly. I mean that the fitted equation based on your results can only be valid in this specific situation. For other researchers, it cannot be used (while the method is applicable). In this case, the demonstration of the equation is not so important. If possible, you can make discussion and recommendations based on your results to show audience the key findings from your results, which could be utilized in other circumstance.

Response to Page 10, Figure 4: A set of plots in a single figure usually requires a compound figure title, where you can describe each plot. Please refer to other papers.

Response to Page 10, Figure 4c: You can edit the plot as a picture. This allows you to wipe A, B, C and insert new letters.

Response to Page 12, “The optimum coded values for the initial metal concentration…which delivered a removal rate of 97.9 %”: Please allow me to clarify my meaning. If you see X2 and X3, you will found they are between -1 and 1, which means the optimal values lie in the range you tested. However, X1 is -1. That is why I think the optimal value is out of the range of test. In this case, I think you need to set a broader range for X1, at least for the lower limit.

Response to Page 14, “Table 5 showed that the R2 values of pseudo-first-order and pseudo-second-order models…”: I am sorry that I cannot find where you made such revision. Did previous studies report that such a process follows second-order kinetics? Please make discussion on this.

Response to Page 16, “It was observed that the adsorption efficiency of our synthesized material was high…”: Have you already shown the results of KAC? If so, please remove Figure 11b to avoid repetition. Such a comparison can be described in text.

Reviewer 3 Report

Journal: Materials

Title: Lignocellulosic based activated carbon loaded silver nanoparticles and chitosan for efficient removal of cadmium and optimization using Response Surface Methodology

Authors: Sujata Mandal 1, Sreekar B Marpu 2, Mohammad A. Omary 2, Catalin C. Dinulescu 3 Victor Prybutok 4 and Sheldon Q Shi *5

The manuscript was re-examined after revision. The revisions and questions for the Manuscript are not fully answered. The novelty of the work is still questionable. Moreover, the graphical representations given within the scope of the study are still not acceptable. After detailed reviews, I do not recommend that the manuscript be published in the journal "materials".

Detailed comments:

The English of the Manuscript is still not sufficient. The entire manuscript should be reviewed.

Novelty of this manuscript still unclear.

Adsorption kinetics still unclear. Why 10 mg/L Cadmimum selected? 10 mg/L can be found in natural conditions?

Table 1 is still visually incorrect. What is the meaning of “and” column means. There is two column named as “coded” and “actual values” but three values shown.

All of figures and figure captions prepared and revised poorly. E.g. Fig 2.’s caption revised as a) and b) but there is no explanation about wavelengths. Moreover, Fig 5a caption stated as “Main effects” what is the conditions? Main effects of what? All of figures are not adequate and unacceptable. Some graphs prepared with Microsoft Excel format but some of them are not. There must be visual harmony between all of the figures. Each graph looks like it was prepared by someone else. In the first comments, it was stated to the authors that this should be corrected, but there is no progress on this issue.

There is still no information or explanation of the temperature in manuscript. Why temperature kept in 27C? Why other temperatures not worked? Still unclear.

Adsorption kinetics; What is the reason for the trend for 10mg/L? What is the difference between the slope obtained at 60 minutes and the slope obtained later? Still unanswered.

The figure is incompatible with the caption of Fig 9. In addition, the results obtained with CAC are almost identical to the results obtained with KAC-Ag. Why should we choose this adsorbent? Its superior features are not understood. Still not answered.

Where are the results of the experimental studies on adsorbent dosage? It is not clear why 20mg was chosen.

After revision Fig 11a has added to the manuscript. But its not understandable and prepared poorly.

Author Response

"Please see the attachment
